# Suppressing Chondrocyte Hypertrophy to Build Better Cartilage

**DOI:** 10.3390/bioengineering10060741

**Published:** 2023-06-20

**Authors:** Christian Shigley, Jay Trivedi, Ozair Meghani, Brett D. Owens, Chathuraka T. Jayasuriya

**Affiliations:** 1The Warren Alpert Medical School, Brown University, Providence, RI 02903, USA; christian_shigley@alumni.brown.edu; 2Department of Orthopaedics, Alpert Medical School of Brown University, Rhode Island Hospital, Providence, RI 02903, USA; jay_trivedi@brown.edu (J.T.); omeghani@gmail.com (O.M.); brett_owens@brown.edu (B.D.O.); 3Division of Sports Surgery, Department of Orthopaedic Surgery, Alpert Medical School of Brown University, Rhode Island Hospital, Providence, RI 02903, USA

**Keywords:** chondrocyte hypertrophy, cartilage restoration, cartilage repair, osteoarthritis

## Abstract

Current clinical strategies for restoring cartilage defects do not adequately consider taking the necessary steps to prevent the formation of hypertrophic tissue at injury sites. Chondrocyte hypertrophy inevitably causes both macroscopic and microscopic level changes in cartilage, resulting in adverse long-term outcomes following attempted restoration. Repairing/restoring articular cartilage while minimizing the risk of hypertrophic neo tissue formation represents an unmet clinical challenge. Previous investigations have extensively identified and characterized the biological mechanisms that regulate cartilage hypertrophy with preclinical studies now beginning to leverage this knowledge to help build better cartilage. In this comprehensive article, we will provide a summary of these biological mechanisms and systematically review the most cutting-edge strategies for circumventing this pathological hallmark of osteoarthritis.

## 1. Introduction

Acute cartilage injury is a leading cause of post-traumatic osteoarthritis (PTOA) [1,2]. After clinical assessment of an acute cartilage injury (i.e., size, location, depth), several options may be considered to attempt cartilage restoration under today’s clinical standards. These options include creation of microfractures in subchondral bone to promote stromal cell migration into the injury site (Figure 1A), implantation of autologous cultured chondrocytes on a porcine collagen membrane (MACI) to reoccupy the defect site (Figure 1B), and the fitted placement of cartilage allograft plugs (or osteochondral grafts) to fill the defect with existing tissue that is expected to be devoid of living cells. Although such approaches have had promising results, these strategies are not without their drawbacks, including induction of fibrocartilaginous scar formation and/or cartilage hypertrophy at the site of injury [3]. Further, global clinical trials currently underway that utilize cells, biological factors, and scaffolds largely do not take into consideration ways to prevent hypertrophic differentiation of cells that form or repopulate cartilage neo-tissue (Appendix A). Chondrocyte hypertrophy is a hallmark of OA [4], and hence its inhibition is critically important to develop more effective therapeutics that combat this disease [5]. This phenotypic change is associated with abnormal tissue homeostasis favoring catabolic proteinase production, formation of pre-osteogenic pericellular matrices, and cell enlargement – all of which is reported to negatively impact the essential architecture of articular joint cartilage [6,7]. 

## 2. The Healthy State of Articular Cartilage

Adult articular joint cartilage (AC) is hypocellular, avascular, and it has an organized dense and permeable extracellular matrix (ECM) that allows for the absorption of nutrients from synovial fluid, hence it does not require an integrated blood supply. AC is smooth and relatively thin with a lubricated surface that minimizes friction and absorbs compressive forces transmitted to the underlying subchondral bone. Its unique viscoelastic properties allow it to withstand recurrent, intense biomechanical loads with little to no tissue damage [8,9]. AC is stratified into zones depending on tissue architecture as well as the phenotype of the cells in each area. The outer most superficial small region is Zone I that consist immature chondrocytes with collagen fibers positioned parallel to the surface. The middle region, Zone II, contains proliferating chondrocytes and collagen fibers that are more obliquely oriented to the articular surface. The deepest region, Zone III, contains some hypertrophic chondrocytes in the calcified cartilage that are found in lacunae that are larger than the previous two zones and oriented in vertical columns, within an ECM that contains type X collagen [10]. Below Zone III is a thin proteoglycan-depleted layer called the tidemark, which separates uncalcified and calcified cartilage. The calcified region of cartilage acts as a transitional interface between hyaline tissue and subchondral bone [11,12]. 

AC ECM is primarily a network of collagens, mostly type II collagen; and proteoglycans, mostly aggrecan [13]. The matrix is also highly charged, which causes the tissue to hold up to 80% water by weight. Additionally, the net negative charge between proteoglycan molecules creates a mutual repulsion to generate an intrinsic stiffness and enables conformational recoil following compression. Chondrocyte-matrix interactions are mediated through mechanical, electrical, and physiochemical signals to balance anabolic and degradative activity [14]. These interactions direct chondrocytes in optimally maintaining homeostasis for sustaining healthy cartilage. Hence, when chondrocytes become dysfunctional or damaged, ECM degeneration follows. 

## 3. Cartilage Hypertrophy in Injury and Catabolic Joint Disease

AC damage caused by acute injury or chronic trauma/disease can manifest as partial or full thickness lesions. Partial thickness lesions do not reach subchondral bone and rely on local chondrocyte proliferation to restore. However, this regenerative response notoriously stops before the defect is completely healed, resulting in reduced tissue integrity [15]. Conversely, full thickness lesions run all the way down to subchondral bone. Bone marrow (BM-) mesenchymal stem cells (MSCs) are capable of migrating into the tissue to differentiate into chondrocytes and remodel the ECM [15]. However, the absence of blood vessels, gradient of physioxia, and the dense ECM of AC acts as barriers to impede significant cell migration and chondrogenesis [16]. In cases of serious traumatic joint injury and/or degenerative joint disease, the AC ECM is damaged to an extent that is disproportional to the natural capacity for it to resynthesize. However, a regenerative response is still initiated as a portion of cartilage cells that have characteristics of mesenchymal progenitor/stem cells will undergo proliferation to resynthesize the damaged tissue [17]. In advanced OA, these cells are suspected to give rise to cellular clusters [18,19], traditionally referred to as “chondrocyte clones” as they eventually reach a state of replicative senescence [20] and enlargement, known as cellular hypertrophy [21,22]. Normally, cellular hypertrophy is an essential part of musculoskeletal development [23]. However, in the setting of severe AC damage or aging, abnormal chondrocyte hypertrophy occurs resulting from disrupted expression of BMPs [24], TGF-β1 [25], and Indian Hedgehog [26], which can be triggered by altered joint loading and inflammation [27,28]. Elevation of collagen type X production by chondrocytes is a strong indicator of chondrocyte hypertrophy. While collagen type X expression is a normal part of early endochondral bone formation, it is minimally expressed in healthy mature AC [29]. Collagen type X expression is reported to be driven by runt-related transcription factor 2 (Runx2) [30], which is ectopically expressed in OA chondrocytes and enhances expression of other OA associated biomarkers, such as alkaline phosphatase (ALP) [31], matrix metalloproteinases (MMPs) [32,33], and vascular endothelial growth factor (VEGF) [34]. Additionally, various studies have documented that the loss of Runx2 in mature AC results in partial reversal of OA phenotype compared to wild type controls [35]. 

Late chondrocyte hypertrophy is characterized by matrix degradation and increased vascularity. These cells can be found throughout the cartilage, including the articular surface, during advanced OA. Matrix degradation is facilitated by increased expression of MMP13 and aggrecanases, such as ADAMTS4. MMP13 is over-expressed in OA cartilage and it is thought to promote disequilibrium from healthy homeostasis to pathologic degradation of collagen type II. Although MMP13 is considered a marker of hypertrophy, it can also be induced through other mechanisms, such as inflammation and mechanical stress [36]. VEGF expression too is a characteristic feature of cellular hypertrophy and OA cartilage. VEGF is responsible for the induction of endothelial cell migration during neo-angiogenesis; however, increased vascularity of cartilaginous tissue is associated with calcification and progression of endochondral ossification [34]. VEGF is also induced by excessive mechanical forces on articular cartilage [37]. Although repeated joint loading is necessary for healthy cartilage ECM, overloading the AC leads to increased VEGF expression in a similar fashion to damage from OA correlating with increased VEGF [38]. Ultimately, the loss of matrix and invasion of endothelial cells promote osteoblast invasion that initiates the cartilage to bone conversion.

Previous studies have noted that hypertrophic chondrocytes share phenotypic similarities to senescent chondrocytes. It has been demonstrated that hypertrophic and senescent chondrocytes co-exist in pathologically stressed OA cartilage tissue. Cellular senescence is stimulated by cellular stressors, including mechanical and oxidative stress, and results in growth arrest, increased cell size, epigenetic methylation, and unique secretory phenotype [39]. OA chondrocytes exhibit features of senescent cells, such as shortened telomeres and increased production of collagen type X, MMP13, and other inflammatory cytokines. These changes promote the production of prostaglandin E2, and nitric oxide that further promote catabolism and inflammation [40]. These molecules result in excessive secretion of proteolytic enzymes that are responsible for cartilage ECM degradation, which further fuels the catabolic activity disequilibrium. 

Hypertrophic chondrocytes directly promote changes of their surrounding matrix, which increases mineralization and reduces the elastic properties of the cartilage and initiates vascular infiltration [41,42]. Hypertrophic chondrocytes produce growth factors and cytokines that recruit chondroclasts to digest the ECM and direct perichondral cells toward an osteoblastic lineage [43]. Then, most hypertrophic chondrocytes undergo apoptosis and create lacunae. The lacunae become space for blood vessels to invade, directed by VEGF signaling, and osteoblasts to lay down bone matrix. At the joint margins, osteophytes arise from cells derived from periosteum and synovium [44]. Additionally, the ongoing changes in the biomechanics and a continues release of cytokines in the subchondral bone may lead to bone cycst formation and sclerosis. 

The inflammatory response to injury plays a critical role in influencing hypertrophic change in cartilage. When healthy chondrocytes are damaged by trauma or overuse, inflammatory cytokines are expressed in the synovial fluid. Sustained release of these factors results in chondrocyte apoptosis, activation of catabolic processes, and ECM disruption. In the setting of chronic, low-grade inflammation, IL-1β,TNF-α, CXCL8, and CXCL1 appear to contribute to OA and correlate with disease severity [45]. As the acute response to injury continues, anabolic processes begin to stimulate chondrocyte proliferation and increase ECM production. However, cytokines, such as IL-1β and TNF-α, are associated with suppressing gene expression by differentiated chondrocytes, such as SOX9, aggrecan, and col II expression, which is critical for chondrogenic differentiation of the cells [46]. Additionally, factors released from damaged subchondral bone matrix, including TGF-β, calcium-phosphate complexes, and exosomal micro RNAs (miR), such as let-7a-5p, drive hypertrophic differentiation of proliferating chondrocytes [47,48,49]. In turn, additional factors secreted from hypertrophic chondrocytes, including VEGF, IL-1β, and TNF-α, promote osteoclastic change and drive further subchondral injury [50]. Ultimately, the accumulation of hypertrophic chondrocytes fails to restore healthy cartilage function, which drives biomechanical alterations of the joint and furthers the damage, thereby continuing the inflammatory response. However, since synovitis starts before structural changes in cartilage, which can be identified by imaging, early intervention to modulate insidious inflammation and aberrant healing remains a difficult challenge [51]. 

## 4. Pathways That Regulate Chondrocyte Hypertrophy

Differentiation of the chondrocytes and hypertrophy are pivotal in bone formation. The long bones of the vertebrate skeleton are formed by endochondral ossification. The ossification process starts with the concentration of MSCs that eventually differentiate to chondrocytes and forms a cartilage template. This newly formed cartilage template is eventually replaced by the bone. The process of MSC condensation and bone formation is governed by cellular signaling. These cells eventually differentiate into chondrocytes with two different populations: (i) circular, low proliferating chondrocytes and (ii) high proliferating chondrocytes that ultimately undergo maturation [52,53,54,55]. During the process of maturation, chondrocytes situated in the central region of cartilage further differentiate into hypertrophic chondrocytes. These differentiated cells produce the mineralized ECM which serves as a template for bone formation [56]. In contrast, the formation of hypertrophic chondrocytes during OA progression is believed to be a result of homeostatic imbalances in the joint microenvironment that favor terminal differentiation and mineralization. Studies have demonstrated that factors inciting this altered state in cells native to articular cartilage tissue include both inflammatory queues and the leaking of growth factors into the cartilage during subchondral bone remodeling (Figure 2) [57,58]. Below are descriptions of documented cellular pathways central to chondrocyte differentiation that are altered in response to homeostatic imbalances, thereby allowing for the initiation of cell hypertrophy. 

### 4.1. Sox9 Signaling

As a member of the “high-mobility group-box” family of transcription factors; Sox9 is essential in chondrocyte differentiation and cartilage development as it is one of the initial markers of chondrocyte condensation. However, the expression of Sox9 is almost completely suppressed in the hypertrophic chondrocytes [59]. Multiple studies have reported that Sox9 plays a pivotal role in chondrogenic differentiation [60]. Heterogenous mutation in the DNA binding site of Sox9 or complete deletion of Sox9 is reported to result in severe skeletal malformation syndrome and blockage of chondrogenesis [61]. Additionally, the deletion of Sox9 also resulted in the differentiation of immature chondrocytes into hypertrophic chondrocytes indicating that Sox9 is essential for the establishment of a chondrogenic lineage [62]. Furthermore, Sox9 regulates the chondrocyte differentiation events through activation of downstream targets that are critical in healthy cartilage homeostasis. Other Sox family member proteins, such as Sox5 and Sox6, can also synergistically regulate chondrocyte differentiation through co-expression of Sox9 [63,64]. Sox9 represses chondrocyte hypertrophy mainly by: (i) inhibiting the transcription factor Runx2 [65], (ii) inhibiting canonical Wnt signaling [65], and (iii) directly repressing expression of hypertrophic chondrocyte markers, such as collagen type X and VEGFA [66]. 

### 4.2. Runx Family Transcription Factors

Runx2 and Runx3 belong to Runx family transcription factors that are regarded as positive regulators of chondrocyte hypertrophy. Various studies till date have demonstrated that the overexpression of Runx2 in undifferentiated chondrocytes results in the elevated expression of several hypertrophy markers including MMP13 and Col 10 [67,68]. Additionally, Runx2 is also a part of transcription complex of the Ihh (Indian Hedgehog) gene to strongly induce its expression. Additionally, Runx2 also interact with BMP-regulated SMAD proteins that ultimately promotes the expression of chondrocyte hypertrophy markers [69,70]. Further, in vivo studies suggest that Runx2 results in delay as well as reduction chondrocyte hypertrophy in mice and deleting both Runx2, and Runx3 results in stunted chondrocyte maturation altogether [69]. These findings collectively suggest that Runx2 and Runx3 are pivotal for chondrocyte maturation and hypertrophy.

### 4.3. Bone Morphogenetic Protein (BMP) Signaling

BMPs are a group of cell signaling molecules that belongs to Transforming Growth Factor beta (TGF-β) superfamily signaling proteins. BMPs were initially identified for their unique ability to induce bone formation [71]. BMPs are reported as positive regulators of the chondrogenesis and ossification processes at different stages [72,73]. Inhibition of BMPs suppress cartilage formation during early stages of limb development [74]. BMP signaling operates through the receptors BMPR1/BMPR2, activation of which results in phosphorylation of SMAD transcription factors. SMAD1, -5, and -8 regulate the expression of downstream target genes associated with differentiation of chondrocyte and hypertrophy [75,76,77]. Inversely, the individual or combined deletion of these receptors result in the lack of expression of Sox5, -6, and -9 in pre-cartilaginous condensations, leading to the defective maturation of chondrocytes. BMP-2, however, has been reported to activate Wnt signaling thereby promoting chondrocyte hypertrophy and cartilage degradation [78]. BMP-7 has been noted to increase chondrogenic potential and suppress the hypertrophic cell phenotype; and BMP-13 has been reported to inhibit osteogenic differentiation in MSCs, thus encouraging differentiation toward a chondrogenic or adipogenic lineage [78,79]. 

Distal-less Homeobox genes (Dlx) 5 and 6 are downstream targets of BMP pathway signaling that regulate skeletogenesis and chondrocyte hypertrophy in association with Sox9 [79,80,81]. During the later stages of skeletogenesis, Dlx5 is highly expressed in prehypertrophic and hypertrophic zones however, Dlx5 is absent in the immature chondroblasts in the resting or the proliferating zone [82]. Previous studies have demonstrated that Dlx5 and Dlx6 are reciprocally expressed during bone formation. These studies were supported by the in vivo findings where deletion and overexpression of either Dlx5 and/or Dlx6 resulted in the reduced proliferation of chondrocytes and increased chondrocyte hypertrophy, respectively [83,84]. Further, Zhu et al. demonstrated that tissue specific misexpression of Dlx5 in transgenic mice promotes chondrocyte hypertrophy, ossification, and abnormal skeletal development [85]. It has been documented that the cartilage-derived mesenchymal progenitors cells (C-PCs) show considerable phenotypic similarities with BM-MSCs, however, the cartilage derived progenitor cells are resistant to the hypertrophic changes [86]. Subsequent studies revealed that the BM-MSCs show significantly higher expressions of Dlx5 as compared to the C-PCs, and knocking down Dlx5 results in the resistance of hypertrophy markers in BM-MSCs [87]. These studies collectively highlight Dlx5 as a potential therapeutic target to prevent hypertrophy and potentially attenuate osteoarthritis. 

### 4.4. Wnt Signaling

Wnt signaling is central for skeletal development. It plays a crucial role in chondrocyte and osteoblast differentiation. Wnt signaling operates through canonical or non-canonical, determined by the utilization of downstream β-catenin [88]. Canonical Wnt signaling relies on the intracellular accumulation of β-catenin and its translocation to the nucleus, while non-canonical signaling functions independently of it [89]. Various studies have demonstrated that the action of β-catenin from canonical Wnt pathway activation promotes osteoblast differentiation while it suppresses the chondrogenic differentiation of MSCs [90,91]. Reinhold et al. [91] reported that Wnt3a strongly inhibits the expression of chondrocyte marker and the process of chondrogenesis. Further, osteoblast precursors without β-catenin expression have been demonstrated to differentiate into chondrocytes [91]. Conversely, noncanonical Wnt signaling regulates the polarity of growing chondrocyte columns in the physis. The disruption of the Wnt planar cell polarity pathway leads to disorganized growth plates in vivo while its activation initiates columnar morphogenesis in vitro [91]. Regarding noncanonical Wnt signaling, Liu et al. [92] reported that Wnt-11 overexpression results in the increased expression of chondrogenic regulatory genes and promotes chondrogenic hypertrophy in synergism with TGF-β. Yang et al. [93] indicated that non-canonical Wnt-5a and Wnt-5b regulate chondrocyte proliferation and hypertrophy via chondrocyte-specific expression of collagen type II. Wnt signaling also plays a regulatory role in the maintenance of polarity of the chondrocytes. Bradley and Drissi [94] demonstrated that Wnt-5b regulated MSC aggregation and chondrocyte differentiation through activation of the Wnt planar cell polarity pathway [95]. Collectively, these findings suggest that canonical Wnt signaling that acts via β-catenin, unlike non-canonical pathway, seems to block chondrocyte differentiation. Modulation of these downstream Wnt targets using Wnt and BMP antagonists is reported to inhibit hypertrophy in chondrogenically differentiated BM-MSCs [96]. 

### 4.5. Indian Hedgehog (Ihh) and Parathyroid Hormone Related Peptide Signaling (PTHrP) Signaling

The Ihh pathway is a necessary regulator of endochondral ossification. Ihh is only expressed in prehypertrophic and early hypertrophic cells. In chondrocytes, Ihh signaling directly activates proliferation, while the absence of Ihh leads to markedly reduced chondrocyte proliferation and premature hypertrophy [97]. Ihh expression is higher in areas of degenerated cartilage, and OA chondrocytes treated with Ihh demonstrated hypertrophic change, indicating a role for Ihh in OA progression [98]. Additionally, Ihh controls the expression of parathyroid hormone-related protein (PTHrP). Overexpression of PTHrP delays chondrocyte differentiation toward hypertrophy, while the deletion of PTHrP diminishes chondrocyte proliferation and ultimately accelerated bone formation [99]. Interestingly, the deletion of PTHrP is sufficient to inhibit chondrocyte hypertrophy despite upregulated Ihh expression [100]. However, in PTHrP knockout mice, cartilage proliferation is still present, although reduced, which suggests a PTHrP-independent pathway could also exist [101]. Additional studies concluded that a PTHrP-independent signaling pathway that operates through the transcription factors of the glioma-associated oncogene (GLI), positively regulates chondrocyte proliferation [102]. 

### 4.6. IκB Kinase/NF-κB and Inflammatory Cytokines 

NF-κB is a rapid-acting transcription factor that is critical for responding to harmful stimuli, including stress, cytokines, free radicals, and foreign antigens. As such, this pathway is particularly linked with inflammatory responses. NF-κB is primarily activated through IκB kinase (IKK) signaling and activation of this pathway leads to chondrocyte hypertrophy [103]. Further, knockdown of IKKα and IKKβ in OA chondrocytes leads to a reversal of the OA phenotype by increasing glycosaminoglycans (GAG) secretion and collagen type II production and reducing calcium deposits [104]. Interestingly, IKKα knockdown only resulted in reduced Runx2 levels while IKKβ knockdown only resulted in increased Sox9 levels. Additionally, inflammatory cytokines downstream of NF-κB, such as TNFα, upregulate hypertrophic markers, reduce GAG expression, and overall erode AC [105]. 

## 5. Pre-Clinical Strategies to Inhibit Chondrocyte Hypertrophy in Joint Cartilage and in Applications of Cartilage Tissue Engineering 

Given our current understanding of the molecular basis of chondrocyte hypertrophy, there is emphasis on using this knowledge to develop strategies for preventing its occurrence in articular cartilage during aging and disease as a preventative OA therapy. Likewise, inhibiting this cell fate is a current focus of tissue engineering applications intended for articular cartilage restoration. This section will discuss promising cells, factors, and molecules that have translational potential for clinical cartilage regeneration. 

### 5.1. Mesenchymal Progenitor Cell Therapy 

Progenitor cells that exhibit MSC characteristics that are advantageous for skeletal tissue repair can be acquired from a multitude of connective tissues, including adipose, cartilage, synovium, ligament/tendon, and bone marrow [106]. BM-MSCs are the most frequently researched progenitor cell type for cartilage repair, either alone or with scaffolds, and they have demonstrated the capacity for stimulating cartilage repair [107,108]. To the best of our knowledge, we only know of one clinical patient study that compared the efficacy of BM-MSCs to other surgical interventions in cartilage repair. The study demonstrated that administering autologous BM-MSCs proved to be at least as effective as ACI in improving patient clinical outcomes for focal chondral lesions in age and lesion size matched cohorts [109]. Additionally, the study demonstrated that BM-MSCs could be implemented in older patients with little consequence on the outcome. The expanded age range of application is an advantage over ACI, which is less successful when used in patients > 45 years old [110]. However, the findings of this study are limited by its small sample size of 72 patients. Moreover, 18 of the patients had received a concomitant procedure, such as ACL repair, patellar realignment, high tibial osteotomy, and partial meniscectomy [111]. 

BM-MSCs though are not without barriers that must be overcome prior to widespread therapeutic implementation. First, cell harvesting from a donor is difficult because they are a rare population with a frequency of 0.01–0.001% in bone marrow [111]. Second, to compensate for low cell count from donor harvest, BM-MSCs are expanded in vitro. However, during the expansion, BM-MSC differentiation potential has been reported to decrease, which has been attributed to shortened telomeres that drive senescent changes [112]. Chondrocyte hypertrophy results in the production of mineralized ECM, which is a template for subsequent remodeling and bone formation. Since these features resemble the endochondral bone formation, it has been suggested that the formation of ectopic endochondral ossification centers is a core characteristic of OA. The hypertrophic tendencies of BM-MSCs during chondrogenesis represent a major challenge to its development for clinical tissue regeneration. 

In addition to BM-MSCs, several other types of stem/progenitor cells are being investigated to inhibit cartilage hypertrophy and promote cartilage regeneration. Due to extensive variability in differentiation potential amongst tissue-specific stem cells, they serve as a great source for generating high-quality neo tissue with minimum potential of hypertrophy [113,114]. For instance, articular chondrocytes are the cells frequently used cell source for ACT. However, their limited availability requires their extensive in vitro expansion before obtaining sufficient cells for implantation [115]. This expansion frequently results in dedifferentiation of the cells which results in the loss of phenotype that promotes hypertrophic changes that ultimately generate cartilage with inferior biochemical properties and increased risks of hypertrophy [116,117]. Articular cartilage progenitor cells that exist in the superficial zone of articular cartilage are considered superior to BM-MSC due to less catabolic changes induced by CPCs as compared to BM-MSC in producing the cartilaginous matrix [86,118]. The investigations with CPCs are expanding rapidly and more clinical evidence will be required the elucidate the mechanistic details of their chondroprotective role. Embryonic stem cells (ECSs) have also served as a potential source to repair damaged tissue because of the infinite ability of proliferation, self-renewal, and differentiation. However, studies have reported hypertrophic differentiation of mouse ECS upon chondrogenic induction in vitro [119]. Due to various legal and ethical restrictions, cell-based therapies require extensive investigation before their active translational implications. 

### 5.2. Growth Factor Therapy

Given the role of TGF-β superfamily to enhance ECM synthesis and differentiation in early cartilage development, researchers have sought to further elucidate qualities of each isoform that can help restore damaged cartilage. Since TGF-β2 is highly expressed in hypertrophic zones of cartilage growth, research relevant to cartilage repair has mostly focused on TGF-β1 and TGF-β3. When comparing TGF-β3 to TGF-β1, TGF-β3 demonstrated stronger chondrogenic potential and rapid differentiation than TGF-β1 [120]. Further, the differentiated chondrocytes treated with TGF-β3 do not demonstrate increased susceptibility to hypertrophy [121]. Mechano growth factor (MGF), a splice variant of IGF-1, has been studied for cartilage regeneration. MGF is a 24-peptide protein that is created from exon 4 and exon 5 of IGF-1. MGF has been shown to regulate MSC migration and promote TGF-β3 expression [122,123,124]. In rabbits MGF successfully recruited BM-MSCs to articular defects and inhibited fibrosis on the cartilage surface to permit cartilage repair. Regenerated cartilage under the effects of MGF creates smooth tissue that is rich in proteoglycans and collagen type II without expression of collagen type I, which indicates fibrocartilage. Song et al. [125] demonstrated that MGF treatment promotes chondrocyte proliferation by inhibiting the apoptosis in OA chondrocytes in rabbits. 

BMPs are critical for osteogenic and chondrogenic differentiation, but BMP-7, in particular, appears to increase chondrogenic potential while preventing hypertrophy, compared to other BMPs [126]. When cultured with AC or BM-MSCs, BMP-7 increases synthesis of collagen type II and mucopolysaccharides [127]. Additionally, cell carrying hydrogel augmented with BMP-7 promotes cartilage and ECM formation [128]. BMP-7 is also established as a protective factor against hypertrophy, and recently, a peptide sequence derived BMP-7 has been identified that attenuates an OA phenotype, including markers of hypertrophy [129]. 

Parathyroid hormones (PTHrP) play a crucial role during the development of the cartilage by maintaining the chondrocytes in a proliferative state and inhibiting their differentiation in to hypertrophic chondrocytes [130]. PTHrP treatment is reported to increase the DNA and GAG content in Mesenchymal stem cells as well as in adipose-derived stem cells, and promotes downregulation of Col10A1 and RUNX2 and upregulating SOX9 and COL2A1. These findings suggest that PTHrP suppresses hypertrophy and promotes chondrogenesis [131]. Various other studies have supported the inhibitory role of PTHrP on chondrocyte hypertrophy [5,132,133]. Additionally, inclusion of PTHrP in the chondrogenic induction medium results in significant down regulation of Col10A1 in cartilage constructs that are engineered from the BM-MSCs of the patients with OA [134]. 

### 5.3. Utilization of Anti-Hypertrophic ECM Proteins

Chondroitin sulfate (CS) is a GAG found in AC and has been widely used in cartilage tissue engineering. Polyethylene glycol (PEG) hydrogels containing CS led to enhanced chondrogenic gene expression and matrix production while downregulating collagen type X, compared to PEG hydrogels alone, suggesting that these chondrocytes resist hypertrophic change [135]. Wu et al. [136] showed that microbeads coated in CS and collagen type II lead to chondrogenic differentiation. However, only CS microbeads retained a pre-hypertrophic state in differentiated chondrocytes, while collagen type II microbeads resulted in hypertrophic maturation of the chondrocytes.

Matrilin-3 is a non-collagenous cartilage ECM protein that is involved in chondrocyte differentiation and skeletal development [137,138,139]. When overexpressed in chondroprogenitor cells, Matrillin-3 led to spontaneous chondrocyte differentiation, based on expression of collagen type II, aggrecan, and GAG content. Matrillin-3 is also essential for TGF-β signaling to limit cartilage hypertrophy [139]. 

### 5.4. Scaffolding

Common synthetic polymers used for cartilage regeneration are polyglycolic acid (PGA), polylactic acid (PLA), and polylactic-co-glycolic acid (PLGA). These polymers demonstrate a range of mechanical properties and have already been clinically implemented in sutures, screws, and pins [140,141]. Of these, only two PLA scaffolds have been clinically used for cartilage repair and have demonstrated improved outcomes for PTOA and focal cartilage defects [142,143]. PLA scaffolds have been able to culture chondrocytes with minimal dedifferentiation and hypertrophy, however results vary depending on molecules and proteins that are infused into the scaffold [144,145]. 

PEG is a nontoxic synthetic polymer that is widely used as a scaffold and hydrogel for chondrocyte delivery. PEG hydrogels promote in vitro and in vivo chondrocyte differentiation [146,147]. Further, Varghese et al [135]. found that PEG hydrogels with chondroitin sulfate enhanced chondrocyte expression profile while down-regulating collagen type X, suggesting that these chondrocytes resist hypertrophic change. PEG can also be combined with natural and synthetic materials to alter its properties. For example, Deng et al. [124] added hyaluronic acid (HA) into physiologically stiff PDLLA-PEG hydrogel to assess the rate of released TGF-β to differentiate MSCs. They found that hydrogel containing HA resulted in more potent MSC differentiation and higher biomechanical strength in vitro and in vivo as compared to conventional TGF-B supplemented medium. 

Scaffolds are also augmented with natural materials, such as alginate, agarose, and collagen. Chitosan is a naturally occurring polysaccharide that exists in abundance in crustacean shells and contains glucosamine and HA. Chitosan has been extensively utilized in cartilage tissue engineering [148,149]. Further studies have demonstrated that chitosan-containing scaffolds stimulate chondrogenic differentiation of MSCs, while also decreasing expression of hypertrophic markers [150,151]. For example, Manferdini et al. [152] differentiated BM-MSCs using PEG scaffolds containing TGF-β3 and 0%, 8%, or 16% chitosan. They found that all scaffolds had similar collagen type 2, SOX9, and aggrecan expression, but only 16% chitosan down-regulated collagen types X and I marker expression.

### 5.5. Culturing Conditions for Cells Used in Cartilage Tissue Engineering and Repair

Therapeutic strategies that rely on expanding cells in vitro prior to their implantation/use may also take oxygen tension into consideration during the culture phase. Oxygen tension in native cartilage has been measured at 1–6%, and additional studies have shown that these hypoxic conditions influence the cartilage phenotype, in addition to cell survival [16]. Hypoxia can drive MSCs toward a chondrocyte lineage and influence dedifferentiated chondrocytes towards a mature chondrocyte phenotype with expression of collagen type II and GAGs [153]. A key pathway to this process is hypoxia-induced factors (HIF), which are cytoplasmic proteins that translocate to the nucleus to mediate cellular responses to hypoxia. Studies of HIF-1α and HIF-1β have demonstrated that hypoxia (2.5% O_2_) promotes MSC differentiation to chondrocytes while normoxia (21% O_2_) increases hypertrophic differentiation that ultimately would ossify [154]. However, HIF-2α expression is crucial in OA pathophysiology, as its increased expression leads to increased cartilage degradation and subchondral sclerosis [155]. 

Additionally, mature chondrocytes are known to lose their ECM-forming phenotype during extended periods of in vitro culture. Since MSCs secrete soluble factors that promote chondrogenic differentiation, these cells can be co-cultured with mature chondrocytes to influence and resist dedifferentiation, in vitro [156]. Co-culturing with mature chondrocytes provides the interactive microenvironment to direct differentiation towards a phenotypically mature cartilage expression profile, reduce hypertrophic expression of collagen type X, and increase ECM secretion by MSCs, in vitro [157,158]. Taken together, these results demonstrate crosstalk between cell subpopulations as mature chondrocytes are stimulated to suppress hypertrophy in MSCs, and MSCs act to maintain secretion of essential ECM molecules by chondrocytes.

## 6. Conclusions

Past and emerging clinical strategies for cartilage repair and resurfacing are geared towards the production of neo tissue that is high in type II collagen and proteoglycans. In their current state, however, these approaches are less equipped to remediate chondrocyte hypertrophy, which unfortunately threatens to detract from their long-term efficacy. Future research efforts for finding effective ways to rebuild damaged cartilage tissue, while actively inhibiting chondrocyte hypertrophy stands to benefit musculoskeletal regenerative medicine. There are many pre-clinical strategies, detailed in this review, that have been reported to help inhibit chondrocyte hypertrophy that includes the use of cells, growth factors, matrix proteins, and even microenvironment conditions. It is imperative that scientists and clinicians who are currently developing powerful cutting-edge technologies that will inevitably set future clinical standards for cartilage injury repair also consider how these technologies can be further tweaked to dissuade chondrocyte hypertrophy, as it is a pathological symptom (if not, a precursor) of OA.

## Figures and Tables

**Figure 1 bioengineering-10-00741-f001:**
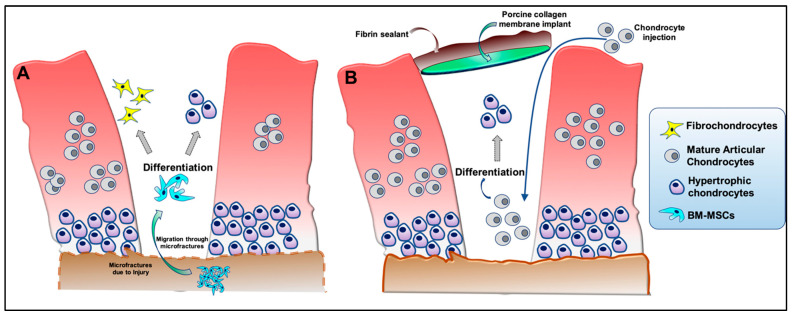
Schematic representation of hypertrophic cell differentiation following the treatment of a cartilage defects with two common standard clinical treatment approaches: (**A**) microfracture surgery/marrow stimulation, and (**B**) the autologous cultured chondrocytes on a porcine collagen membrane (MACI) procedure.

**Figure 2 bioengineering-10-00741-f002:**
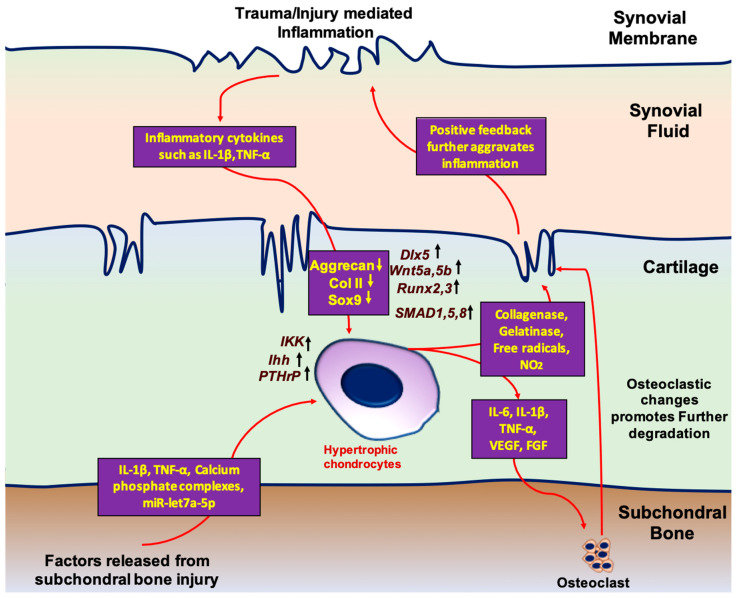
Schematic representation of critical molecular events and feedback mechanisms regulating chondrocyte hypertrophy during OA.

## Data Availability

Not applicable.

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
