# Peer review of "Suppressing Chondrocyte Hypertrophy to Build Better Cartilage"

_bioengineering, 2023, doi:10.3390/bioengineering10060741_

Round 1
Reviewer 1 Report
Abstract OK
2.0 please precise adult articular cartilage
please use catabolic activity instead degradative
3.0 please note that there is a gradient of physioxia in the hyaline cartilage. Please precise TGFß type. line 142: especially in traumatic conditions. Inflammatory cytokines come mainly from OA-associated synovitis and subchondral edema.
4.3: italics?
5.5: Could it be possible to write one or two paragraphs on experimental in vivo data, and especially the difference on heterotopic (subcutaneous) and homotropic intrachondral grafts in termes of hypertrophy and calcification. Gene therapy opportunities should be deveopped, as intra articular drug vectorization.
Refs OK
Figs OK
Author Response
We thank reviewers and the editorial team for their evaluation and comments of our manuscript. We have made changes as suggested by the reviewers and have revised the manuscript. Please find our response to reviewer’s comments below.
REVIEWER #1
Abstract OK
2.0 please precise adult articular cartilage
- Thank you for the suggestion. We have made changes as suggested.
please use catabolic activity instead degradative
- Degenerative is changed to catabolic activity as suggested.
3.0 please note that there is a gradient of physioxia in the hyaline cartilage.
-Thank you for the suggestion. Appropriate changes have been made in the revised manuscript. Please see lines 84-86 in revised manuscript.
Please precise TGFß type.
- TGFß is corrected to TGFß-1 in the revised manuscript.
line 142: especially in traumatic conditions. Inflammatory cytokines come mainly from OA-associated synovitis and subchondral edema.
- Appropriate changes with citations are incorporated in the revised manuscript.
4.3: italics?
- Thank you for pointing out. The revised manuscript is formatted consistently throughout.
5.5: Could it be possible to write one or two paragraphs on experimental in vivo data, and especially the difference on heterotopic (subcutaneous) and homotropic intrachondral grafts in termes of hypertrophy and calcification. Gene therapy opportunities should be deveopped, as intra articular drug vectorization.
- We have discussed individual studies in their respective sections with appropriate citations. Hence, we believe that adding a separate section might lead to redundancy in the texts. Also, we were unable to find substantial literature concerning the differences between grafts mentioned by the reviewer with regards to hypertrophy and calcification. And yes, gene therapy strategies for altering hypertrophy pathways would be an excellent step forward.
Refs OK
- Thank you.
Figs OK
- Thank you.
Reviewer 2 Report
The authors provided a well-structured review on an interesting topic related to the prevention of cartilage hypertrophy. The review presents the mechanisms of hypertrophy development and trends in the suppression of this phenomenon. I would recommend that the authors, in addition to the diagrams presented in the review, add figures from original studies for a more accurate understanding of the topic by readers. In addition, I believe that it is necessary to add a table that summarizes the main studies on the mechanisms of prevention of hypertrophy and clinical studies.
In general, the review covers a wide range of articles, but does not link to a related review (Pharmaceutics 13(8):1139 DOI:10.3390/pharmaceutics13081139), as well as an earlier review (Genes & Diseases Volume 2, Issue 1, 2015, 76-95 DOI:10.1016/j.gendis.2014.12.003)
Author Response
We thank reviewers and the editorial team for their evaluation and comments of our manuscript. We have made changes as suggested by the reviewers and have revised the manuscript. Please find our response to reviewer’s comments below.
REVIEWER #2
The authors provided a well-structured review on an interesting topic related to the prevention of cartilage hypertrophy. The review presents the mechanisms of hypertrophy development and trends in the suppression of this phenomenon. I would recommend that the authors, in addition to the diagrams presented in the review, add figures from original studies for a more accurate understanding of the topic by readers.
- We thank the reviewer for their suggestion. We believe that including figures from original studies will (1) substantially increase the number of figures in the manuscript and (2) We may not be able to acquire all necessary images from the original studies due to copyrights and reprinting rights associated with the studies/article of respective journals.
In addition, I believe that it is necessary to add a table that summarizes the main studies on the mechanisms of prevention of hypertrophy and clinical studies.
- We thank reviewer for the suggestion. We have included a table with clinical studies in the supplementary material.
In general, the review covers a wide range of articles, but does not link to a related review (Pharmaceutics 13(8):1139 DOI:10.3390/pharmaceutics13081139), as well as an earlier review (Genes & Diseases Volume 2, Issue 1, 2015, 76-95 DOI:10.1016/j.gendis.2014.12.003)
- We thank reviewer for their suggestion of these valuable references. As suggested, the citations are added in the revised manuscript at appropriate places.
Reviewer 3 Report
Chondrocyte hypertrophy is a problem during and after repair of damaged cartilage tissue, to minimize the risk of hypertrophic neo tissue formation represents an unmet clinical challenge.
In this manuscript, the author firstly introduces the structure of normal cartilage tissue, and then reviews in detail the etiology of chondrocyte hypertrophy after cartilage damage and the molecular mechanisms regulating chondrocyte hypertrophy. Finally, some current preclinical strategies for inhibiting chondrocyte hypertrophy in articular cartilage and their application in cartilage tissue engineering are summarized. This is an interesting review article, the manuscript writing is well, and easy to read.
I have no comment except for some writing errors that need to be carefully checked and corrected.
Author Response
We thank reviewers and the editorial team for their evaluation and comments of our manuscript. We have made changes as suggested by the reviewers and have revised the manuscript. Please find our response to reviewer’s comments below.
Chondrocyte hypertrophy is a problem during and after repair of damaged cartilage tissue, to minimize the risk of hypertrophic neo tissue formation represents an unmet clinical challenge.
In this manuscript, the author firstly introduces the structure of normal cartilage tissue, and then reviews in detail the etiology of chondrocyte hypertrophy after cartilage damage and the molecular mechanisms regulating chondrocyte hypertrophy. Finally, some current preclinical strategies for inhibiting chondrocyte hypertrophy in articular cartilage and their application in cartilage tissue engineering are summarized. This is an interesting review article, the manuscript writing is well, and easy to read.
I have no comment except for some writing errors that need to be carefully checked and corrected.
- We thank the reviewer for the time and evaluation. We have fixed the writing errors and have made changes in the revised manuscript.